# Epigenetic Findings in Twins with Esophageal Atresia

**DOI:** 10.3390/genes14091822

**Published:** 2023-09-20

**Authors:** Michal Błoch, Piotr Gasperowicz, Sylwester Gerus, Katarzyna Rasiewicz, Arleta Lebioda, Pawel Skiba, Rafal Płoski, Dariusz Patkowski, Pawel Karpiński, Robert Śmigiel

**Affiliations:** 1Department of Family and Pediatric Nursing, Wroclaw Medical University, 51-618 Wroclaw, Poland; michal.bloch@formuladobra.pl; 2Department of Medical Genetics, Medical University of Warsaw, 04-768 Warsaw, Poland; 3Department of Pediatric Surgery and Urology, Medical University of Wroclaw, 51-618 Wroclaw, Poland; sylwester.gerus@umw.edu.pl (S.G.);; 4Division of Molecular Techniques, Department of Forensic Medicine, Wroclaw Medical University, 51-618 Wroclaw, Poland; 5Department of Genetics, Wroclaw Medical University, 51-618 Wroclaw, Poland; 6Department of Pediatrics, Endocrinology, Diabetology and Metabolic Diseases, Medical University of Wroclaw, 51-618 Wroclaw, Poland

**Keywords:** esophageal atresia, Rho signaling pathway, epigenetics, methylation, rare disease, genomics

## Abstract

Esophageal atresia (EA) is the most common malformation of the upper gastrointestinal tract. The estimated incidence of EA is 1 in 3500 births. EA is more frequently observed in boys and in twins. The exact cause of isolated EA remains unknown; a multifactorial etiology, including epigenetic gene expression modifications, is considered. The study included six pairs of twins (three pairs of monozygotic twins and three pairs of dizygotic twins) in which one child was born with EA as an isolated defect, while the other twin was healthy. DNA samples were obtained from the blood and esophageal tissue of the child with EA as well as from the blood of the healthy twin. The reduced representation bisulfite sequencing (RRBS) technique was employed for a whole-genome methylation analysis. The analyses focused on comparing the CpG island methylation profiles between patients with EA and their healthy siblings. Hypermethylation in the promoters of 219 genes and hypomethylation in the promoters of 78 genes were observed. A pathway enrichment analysis revealed the statistically significant differences in methylation profile of 10 hypermethylated genes in the Rho GTPase pathway, previously undescribed in the field of EA (*ARHGAP36, ARHGAP4, ARHGAP6, ARHGEF6, ARHGEF9, FGD1, GDI1, MCF2, OCRL*, and *STARD8*).

## 1. Introduction

Esophageal atresia (EA) with or without tracheoesophageal fistula (TEF) (MIM 189960 and ORPHA 88893), is the most common malformation of the upper gastrointestinal tract, with an incidence of 1:3500 [1]. The frequency varies by region of the world [2]. There are several classifications of EA; the most used is the Gross classification, which distinguishes six types of defects (A–F) [3]. The most common type is Gross type C: EA with concomitant distal TEF—approx. 86% [4]. Clinically, esophageal atresia may occur as an isolated defect or as part of syndrome. Nowadays, patients with EA/TEF are operated on by the thoracoscopic approach, with increasing success [5]

EA is one of the symptoms found in over 50 different diseases and syndromes [6,7]. While syndrome forms with additional congenital anomalies in the cardiovascular, musculoskeletal, urinary, gastrointestinal, or central nervous system are quite well-understood, as for the cause of their occurrence, approx. 50% of cases are isolated defects of unknown etiology. 

Congenital anomalies are more common among twins, and EA is up to 3.2 times more common among twins than in the general population [8]. The occurrence of an isolated form of EA in both twins or in siblings are show in case reports [9,10]. Twins during fetal development are exposed to the same environmental conditions—potentially teratogenic substances and infections—so the occurrence of a defect may be related to the individual susceptibility of a given fetus or the genetic factors of a particular person [11].

Embryologically, the esophagus begins differentiation from the foregut in the fourth week of fetal life, during which time the action of a teratogenic factor or a disturbance of one of the signaling pathways leads to the occurrence of EA. The course of the embryogenesis processes’ differentiation of the trachea and esophagus from the foregut is the subject of research conducted on animal embryos [12]. Four models of how the trachea emerges from the foregut were already described: the outgrowth model, watershed model, septation model, and splitting and elongation model [12,13].

The cause of EA formation remains unclear. Due to the low risk (population risk) of repeating the isolated form of EA in another child and the low risk of EA in the next generation, a multifactorial etiology is postulated: individual sensitivity and the interaction of genetic and environmental factors lead to EA [14]. So far there are no unequivocal reports in the literature regarding the risk factors for the occurrence of EA in an isolated form.

## 2. Materials and Methods

### 2.1. Patients

The patients were six pairs of twins, in which one child was born with EA with distal TEF (Gross type C) as an isolated defect while the other twin was healthy. There were five pairs of same-sex twins, including two pairs of girls and three pairs of boys, and one pair of opposite-sex twins; three pairs were monozygotic, and three pairs of dizygotic twins were confirmed by microsatellite analysis using 13 markers. Details about each pair of twins are shown in the Table 1. 

The analysis was carried out on 16 DNA samples including 6 whole-blood samples obtained from EA patients, 6 whole-blood control samples obtained from healthy twins of EA patients, and 4 esophagus samples obtained from EA patients during surgical thoracoscopic procedure—all samples were collected during the neonatal period or infancy. In addition, individuals with EA were analyzed by CGH array (Agilent SurePrint G3 CGH 8 × 60 k) to exclude confounding effect of chromosomal aberrations. 

### 2.2. Methods

The collection of a specimen of the lower segment of the esophagus by a surgeon took place during the primary or secondary corrective surgery for EA. The tissue sample was immediately flushed in NaCl by complete immersion after collection and then fully submerged in RNALater stabilizer. After immersion, it was stored in a PCV container in a frozen state at −80 °C temperature. 

DNA isolation was performed after thawing the tissues using the QIAamp DNA Mini Kit according to the procedures recommended by the manufacturer (Qiagen). Genome wide methylation was studied by use of reduced representation bisulphite sequencing (RRBS) technique. In brief, isolated DNA was modified by Zymo EZ DNA Lightning Kit (Zymo Research) and paired-end sequenced (2 × 100 bp) on HiSeq 2500 Sequencer (Illumina) in six batches. RRBS libraries were prepared using NEXTflex Bisulfite-Seq Library Prep Kit (Perkin Elmer).

### 2.3. Data Processing

Raw data obtained from RRBS were initially processed using the RTA program provided by the manufacturer, followed by demultiplexing using CASAVA software. Subsequently, adapter sequences were removed using Cutadapt software in the Trim Galore script [15]. Qualitative analysis of raw data, as well as preliminary processing (mapping to the GRCh37/hg19 human genome version, filtering polymorphic variants) and calculation of differential methylation between the studied groups, was performed using the annotatr, genomation, and methylKit packages [16,17]. Only nucleotides present in all samples and with a coverage of at least 10 reads at a given position were included in the analysis. Polymorphic variants were filtered using data from the NCBI dbSNP database (Build 150). Prior to differential methylation (DM) analysis, we summarized methylation information over CpG islands. Mapping to CpG islands was performed using the CpG island definition developed by Wu et al. [18]. 

### 2.4. Statistical Analyses

All calculations were performed in the R/Bioconductor environment. For comparisons between groups, a statistically significant difference in methylation for a given cytosine or region (e.g., CpG islands) was considered if it exhibited a minimum of 20% methylation and had a q-value ≤ 0.01 (q-value is equivalent to a *p*-value adjusted for multiple comparisons) calculated by logistic regression. Calculations were adjusted for possible confounders such as age and gender. Methylome visualizations were generated based on self-organizing maps using the oposSOM package [19]. Unsupervised clustering was performed using the NMF package with hierarchical clustering. Principal component analysis (PCA) was conducted using the factoextra package. Result visualizations were created using the methylKit, factoextra, and NMF packages [20]. Pathway enrichment analysis was performed using the WebGestalt program with pathway definitions from the Reactome database [21]. Significantly enriched pathways were defined as those with a corrected *p*-value of ≤0.05 (FDR ≤ 0.05).

## 3. Results

### 3.1. Quality Check and Filtering

First, for each sample, we checked the QC stats for methylation data such as the coverage and percentage of methylation distribution. A qualitative analysis of reads for each sample was conducted prior to the filtering stage. This analysis was performed using histograms illustrating the distribution of methylation levels (Appendix A) as well as separate histograms depicting coverage (Appendix A). All samples met the quality requirements. Next, each sample was filtered based on coverage to mitigate possible PCR biases by discarding bases with a very high read coverage. 

After merging the data for all samples, nucleotide unification, and read-based filtering, a matrix encompassing the methylation status of 3,169,179 CpG dinucleotides was obtained. Following the filtration of dinucleotides located in polymorphic positions, 3,094,244 CpG dinucleotides remained. Subsequently, the annotation (assignment) of individual CpG dinucleotides to different CpG island elements and gene features was performed. In terms of CpG island structure and location, the majority of CpG dinucleotides (50%) were in intergenic CpG islands, followed by CpG islands associated with genes (30%), CpG shores (14%), and CpG shelves (3%). Regarding gene structure, the highest number of CpG dinucleotides were in introns and intergenic regions (totaling 55%), followed by exons (14%) and promoters (9%). For the statistical analyses, methylation data collected from 946,317 CpG dinucleotides located within CpG islands were utilized, considering their potential impact on the transcriptome and, consequently, tissue phenotype.

### 3.2. Unsupervised Analyses

Initially, methylome visualizations were performed based on CpG island methylation levels using self-organizing maps (Figure 1). 

The analysis of the generated maps revealed significant heterogeneity among the samples within each examined tissue (esophagus, blood). The unsupervised clustering of the maps using a phylogenetic tree is shown in Figure 2. 

This indicates that the EA-associated esophagus samples exhibited distinct methylation patterns compared to blood samples. Furthermore, at this stage, it was not possible to differentiate between blood samples from individuals with or without EA, since both types of samples cluster closely together. Outlying samples can be identified as “ctrl_blood1” and “atresia_blood2”. In the case of samples isolated from the esophagus, a clear high variation in methylome maps can be observed (Figure 1C), which is reflected in the substantial distances between these samples on the phylogenetic tree (Figure 2).

Since the above results were based on an intermediate analysis of self-organizing methylation portraits, it was decided to perform direct clustering at the CpG dinucleotide methylation levels using hierarchical clustering (Figure 3). 

An analysis of the hierarchical tree revealed the division of samples into two main groups. One group consisted of the samples “atresia_blood_2”, “atresia_esoph_3”, “atresia_esoph_5”, and “ctrl_blood_1”. The second group was composed of the remaining samples. An additional analysis of sample grouping using PCA showed the presence of several outlier samples (“atresia_blood_2”, “atresia_esoph_3”, “atresia_esoph_5”, and “ctrl_blood_1”) and the remaining relatively cohesive group (Figure 4). 

Since the variability of the X and Y chromosomes between sexes and the epigenetic silencing of gene expression on the inactive X chromosome are important factors that can interfere with DNA methylation studies, an additional analysis of sample grouping was performed using hierarchical clustering, with the exclusion of the CpG dinucleotides located on the X and Y chromosomes (Figure 5). The analysis did not reveal significantly different clustering compared to the clustering obtained without excluding the X and Y chromosome locations. 

In summary, significantly distinct DNA methylation profiles were observed in samples isolated from the esophagus. Due to the high variability/heterogeneity of the methylome in these samples, they were excluded from further analysis. Additionally, the recurring occurrence of two outlier samples isolated from blood, labeled as “ctrl1” and “atresia2”, was observed, and they were also excluded from further analyses as outliers.

### 3.3. Differential Methylation (DM) Analysis

We focused on the comparison of the DNA methylation in the whole blood of EA patients vs. control samples. To identify significant differences in the methylome, data based on 946,317 CpG dinucleotides located within CpG islands were used. The methylation values were averaged for each of the 65,699 investigated CpG islands. Furthermore, the potential existence of other confounding factors related to sample characteristics, such as batch number, age, and gender, was analyzed. Gender was identified as a variable that could potentially confound the supervised analysis. Subsequently, all calculations were performed with a gender adjustment. 

As result of the differential methylation (DM) analysis, 2056 hypermethylated and 3399 hypomethylated CpG islands were identified in individuals with EA compared to the blood methylome obtained from the control group. The majority of hypermethylated CpG islands were found in introns (451), followed by gene exons (387), gene promoters (305), and 3’ UTR sequences, which had the fewest (28). The highest number of hypomethylated islands was observed in introns (1119), followed by intergenic regions (587), gene exons (467), gene promoters (228), and 3’ UTR sequences, which had the fewest (125). Figure 6 depicts the chromosomal localization of 5455 CpG islands exhibiting significantly different methylation patterns in individuals with EA. The majority of hypermethylated islands were located on the X chromosome, while the majority of hypomethylated islands were located on autosomes. 

Additionally, the localization of hypermethylated CpG island regions and hypomethylated CpG island regions associated with genes relative to the transcription start site (TSS) was assessed. As shown in Figure 7, nearly 60% of hypermethylation events were found in proximity to the TSS. 

Due to the potential effect of CpG island methylation in gene promoters on transcription, subsequent enrichment analyses were limited to CpG islands located in the promoters of known genes. In total, the hypermethylation of 249 CpG islands in the promoters of 219 genes and the hypomethylation of 81 islands in the promoters of 78 genes were identified (Appendix A). 

A pathway enrichment analysis revealed a statistically significant involvement of 10 hypermethylated genes in the Rho GTPase pathway (FDR-adjusted *p*-value = 0.004; Table 2 and Figure 8). However, no statistically enriched pathway was found among the hypomethylated genes. 

No significant differences in hypermethylation or hypomethylation were found within the promoters of genes described in the literature as being potentially involved in the pathogenesis of esophageal atresia, based on Brosens et al., 2021, and Edwards et al., 2021 [6,7].

## 4. Discussion

The etiology of isolated EA, despite numerous previous attempts to identify its causes, remains unknown. There is no certainty regarding the genetic, epigenetic, or environmental factors that influence the increased risk of isolated EA occurrence. For our study, we decided to use the methylation analysis technique, RRBS, which offers the highest resolution in addition to whole-genome sequencing (WGBS). This technique provided access to the analysis of over 3 million CpG dinucleotides located in various genomic elements and/or genes. The analyses in this study, particularly those related to unsupervised and supervised statistical analysis, were limited to 946,317 CpG dinucleotides located in CpG islands. This decision was motivated by the fact that among the known genomic elements, CpG island methylation has the strongest association with gene expression regulation [22]. Therefore, it is within this group of CpG dinucleotides that methylation variability could potentially play a role in EA etiology by affecting gene expression disorders.

As a result of unsupervised analyses, using self-organizing maps of methylation and hierarchical clustering as well as PCA, significant heterogeneity was observed within the methylation profiles of the four esophageal samples. This heterogeneity manifested as distinct methylation maps with a low number of shared features, relatively large distances between the esophageal samples on the phylogenetic tree, and PCA visualization [23]. Heterogeneity within a single tissue’s methylation profile is a well-described phenomenon in the literature and may result from differences in the cellular composition of the sampled tissues or contamination by other cell types, such as blood leukocytes [24]. In the case of studies based on a small number of samples, high heterogeneity is a highly unfavorable phenomenon that significantly disrupts inference, especially in the case of supervised methods, which assume the relative homogeneity of the compared groups [25].

It is also worth noting the large distances between the esophageal samples and the blood samples on the phylogenetic tree and PCA. This indicates significant differences in the CpG dinucleotide methylation profiles resulting from tissue-specific epigenetic changes [26]. Therefore, comparing methylation profiles between the esophageal tissues of EA patients and the methylation profiles in the blood of healthy individuals would primarily reveal methylation changes resulting from natural tissue differences, thereby obscuring differences related to the presence or absence of EA. Based on the arguments mentioned above, esophageal samples were excluded from further comparative analysis in this study.

As indicated by the unsupervised analyses, the blood samples exhibited relatively high homogeneity, forming a tight cluster in the PCA visualization. Summarizing the methylation data into CpG islands and eliminating the esophageal samples and outliers revealed a new pattern of clustering clearly based on EA status, with the EA samples separating from the control samples in both hierarchical clustering and PCA. This indicates that both subgroups were characterized by different CpG dinucleotide methylation patterns. Considering that the methylation profile of whole blood is strongly influenced by the proportions of different leukocyte populations, this may suggest that EA patients had a different blood cell composition compared to the control group [27]. 

Comparing the methylation profiles in blood between individuals with EA and the control group revealed statistically significant differences in the hyper- or hypomethylation of CpG islands in the promoters of several hundred genes. The pathway enrichment analysis revealed the statistically significant involvement of 10 hypermethylated genes in the Rho GTPase pathway (*ARHGAP36*, *ARHGAP4*, *ARHGAP6*, *ARHGEF6*, *ARHGEF9*, *FGD1*, *GDI1*, *MCF2*, *OCRL*, and *STARD8*). The Rho-GTPase pathway is involved in many biological processes [28]. As early as the beginning of the 21st century, the involvement of genes belonging to the Rho GTPase pathway in epithelial morphogenesis was described [29]. Genes involved in the Rho GTPase pathway play multiple roles in tissues, including the regulation of microtubule and actin cytoskeleton dynamics, thereby impacting cell adhesion, migration, and polarity [30]. Consequently, alterations in gene expression within the Rho GTPase pathway are often observed in tumors, and these changes affect the metastatic process of cancers [31]. Importantly, Rho GTPases display a strong link to the regulation of cytoskeletal dynamics, cell polarity, and the trafficking and proliferation of immune cells. Consequently, mutations in Rho GTPases are increasingly recognized to be involved in severe human diseases, often in chronic and life-threatening immune syndromes [32]. To date, no studies linking the Rho GTPase pathway to EA have been published. The involvement of Rho GTPase pathway disturbances in embryopathological processes is likely. In 2021, Zhou et al. discovered that reduced expression of the *CDC42* gene, belonging to the Rho GTPase family, was observed in patients with biliary atresia (OMIM 210500), which may contribute to the etiology of this congenital defect [33]. Therefore, the hypermethylation of the promoters of 10 genes within the Rho GTPase pathway, leading to decreased gene expression, may increase the risk of EA in newborns. Furthermore, the involvement of Rho GTPases in immune regulation may suggest that the pathogenesis of EA may be linked to aberrant inflammatory responses, but it is also possible that the methylation changes are linked to the effects of EA; thus, further research is needed to obtain more evidence for this hypothesis. This discovery is a novel aspect in the context of EA and requires further research, including the analysis of transcriptomes and methylomes from samples isolated from the esophagus. However, due to the very limited access to esophageal tissue, especially from control, healthy individuals, an analysis of plasma cell-free DNA methylation could provide an opportunity to study EA in much larger sample sizes. The verification of the results presented in this study relies on finding significant correlations between the levels of the CpG promoter methylation of the genes and their expression. 

Considering the predominance of the male sex in the occurrence of esophageal atresia, with a ratio to females of approximately 3:2 [14,34], it can be presumed that the susceptibility to the condition is influenced by genetic factors, including the differential functioning of genes located on sex chromosomes. The influence of processes such as the selective gene methylation on the X chromosome in females, known as X-chromosome inactivation, on this process cannot be excluded. Further research is needed to investigate this issue in the population of children with esophageal atresia. Due to the limited sample size, it was not possible to separately profile women and men using comparative analyses, which precludes drawing conclusions from the obtained methylomes in this regard.

## Figures and Tables

**Figure 1 genes-14-01822-f001:**
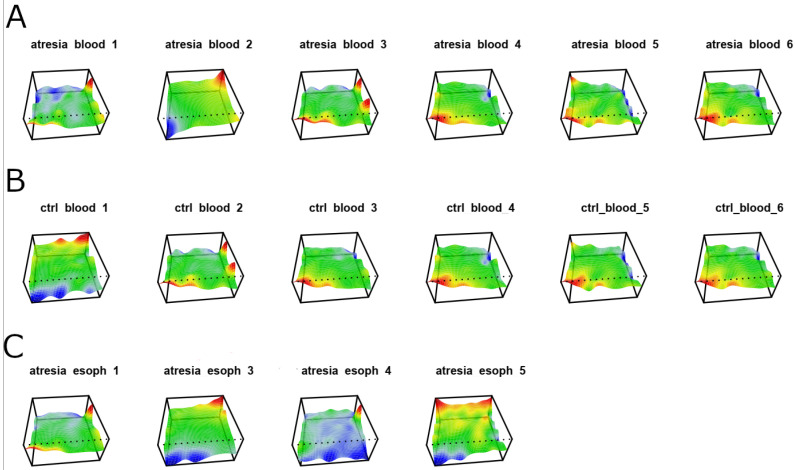
Methylome maps of individual samples were generated based on the self-organization of 946,317 CpG dinucleotides located within CpG islands. The color corresponds to the levels of methylation in CpG dinucleotide groups, with red representing high methylation, yellow and green representing intermediate methylation, and blue representing low methylation. (**A**) Samples isolated from the blood of individuals with EA are described as “atresia blood”. (**B**) Samples isolated from the blood of individuals without esophageal stenosis are described as “ctrl blood”. (**C**) Samples isolated from the EA esophagus are described as “atresia esoph”.

**Figure 2 genes-14-01822-f002:**
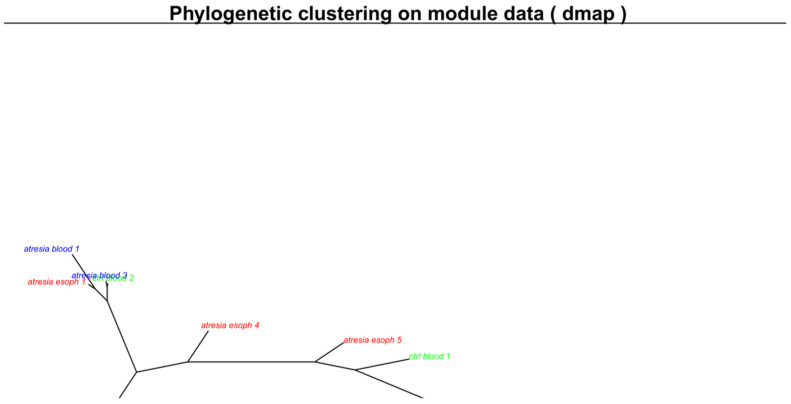
Unsupervised clustering of methylome maps presented in Figure 1 using a phylogenetic clustering by neighbor joining algorithm. Samples located closely together exhibit similar maps, while samples that are farther apart have differing maps. The respective sample groups are indicated by color: blue—samples isolated from the blood of individuals with EA are described as “atresia blood”; green—samples isolated from the blood of individuals without EA are described as “ctrl blood”; red—samples isolated from the EA esophagus are described as “atresia esoph”.

**Figure 3 genes-14-01822-f003:**
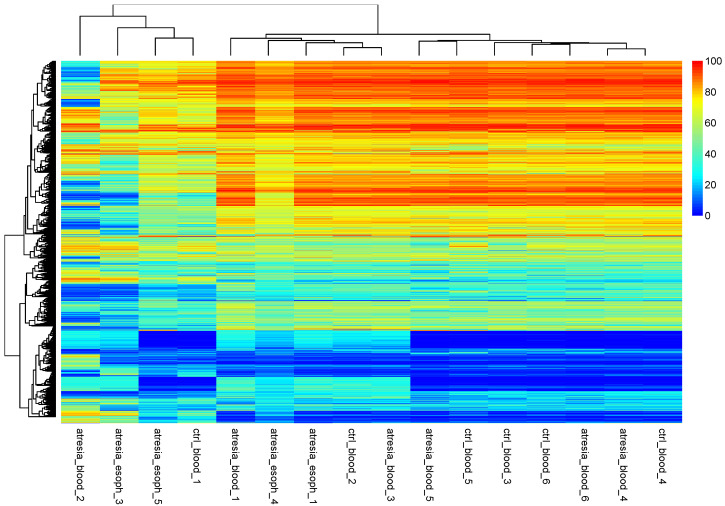
Hierarchical clustering performed at the methylation levels of individual CpG dinucleotides. The observation of the hierarchical tree indicates that the samples can be divided into two main groups. One smaller group consists of the samples “atresia_blood_2”, “atresia_esoph_3”, “atresia_esoph_5”, and “ctrl_blood_1”. The second group is composed of the remaining samples.

**Figure 4 genes-14-01822-f004:**
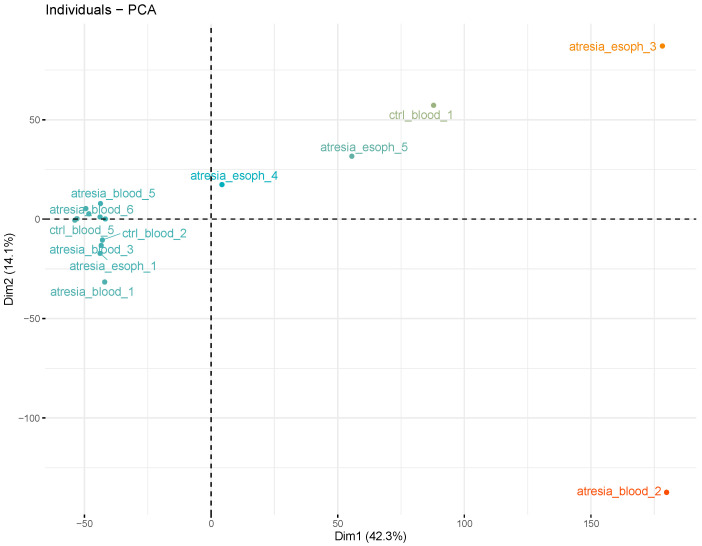
Analysis of sample grouping using principal component analysis (PCA), performed at the methylation levels of individual CpG dinucleotides. This reveals the presence of several outlier samples (“atresia_blood_2”, “atresia_esoph_3”, “atresia_esoph_5”, and “ctrl_blood_1”) and the remaining compact group.

**Figure 5 genes-14-01822-f005:**
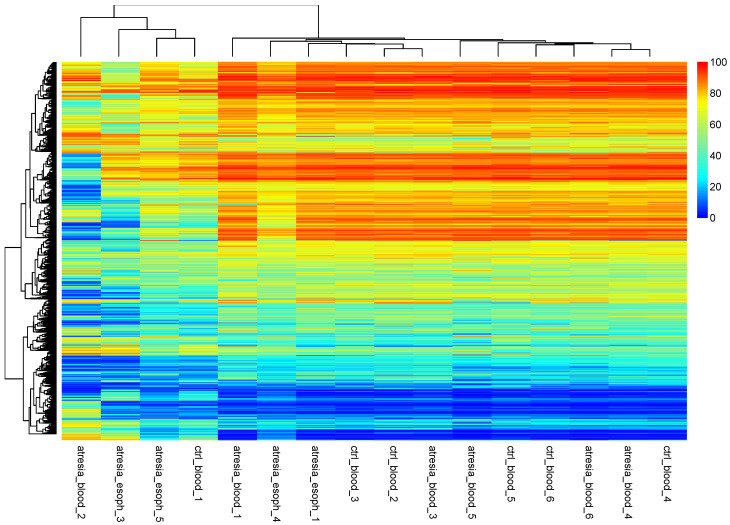
Hierarchical clustering performed at the methylation levels of individual CpG dinucleotides, excluding the X and Y chromosomes. The observation of the hierarchical tree indicates that the samples can be divided into two main groups. One smaller group consists of the samples “atresia_blood_2”, “atresia_esoph_3”, “atresia_esoph_5”, and “ctrl_blood_1”. The second group is composed of the remaining samples. A similar arrangement was obtained in the analysis presented in Figure 3.

**Figure 6 genes-14-01822-f006:**
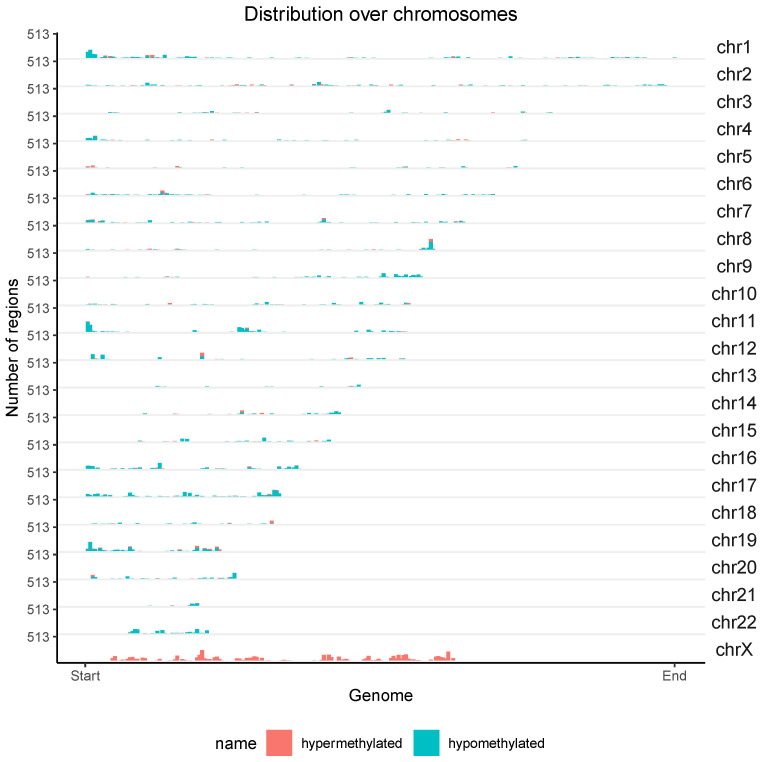
Chromosomal distribution of 5455 CpG islands with significantly different methylation levels in the blood of individuals with EA compared to the methylation levels observed in the blood of the control group. Red indicates hypermethylation, while green represents hypomethylation. The Y-axis denotes the chromosome numbers.

**Figure 7 genes-14-01822-f007:**
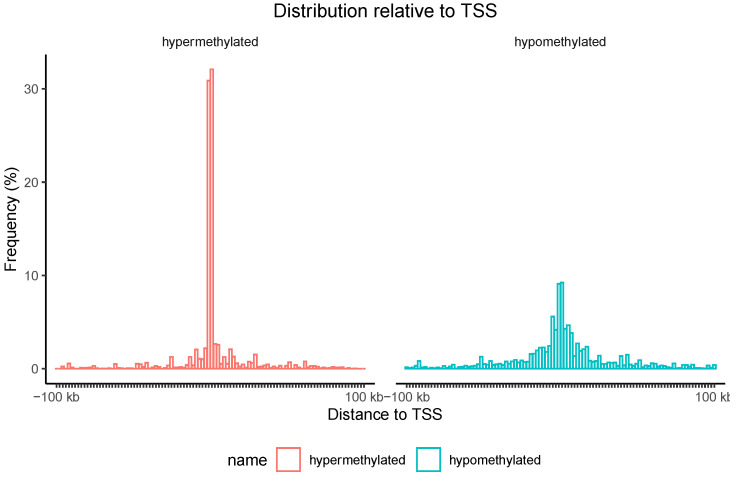
The distribution of CpG islands associated with genes relative to the transcription start site (TSS) with significantly different methylation levels in the blood of individuals with esophageal atresia (EA) compared to individuals in the control group. Red represents hypermethylation, while green denotes hypomethylation. The Y-axis indicates the percentage of islands (%).

**Figure 8 genes-14-01822-f008:**
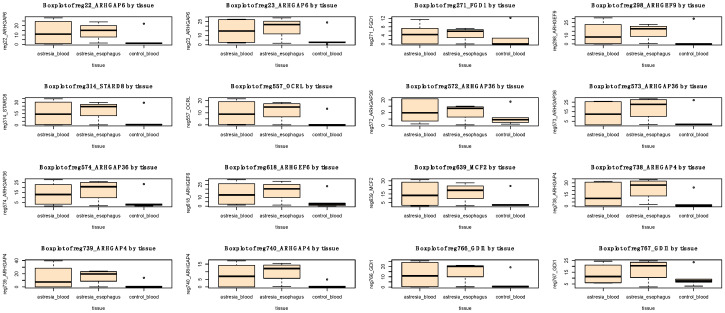
Boxplots representing methylation levels of 16 significantly hypermethylated regions (in blood of individuals with EA in comparison to blood of control group) located in 10 Rho GTPase pathway genes. In addition, methylation levels in esophageal tissues are also depicted.

**Table 1 genes-14-01822-t001:** Details of the six pairs of twins included in each pair: patient with EA (a), healthy child from control group (b) and their zygosity, gender, and EA type. If esophagus tissue sample was collected, it is marked (t).

	1a	1b	2a	2b	3a	3b	4a	4b	5a	5b	6a	6b
Zygostity	Dizygotic		Monozygotic		Dizygotic		Monozygotic		Monozygotic		Dizygotic	
Gender	F	F	F	F	F	M	M	M	M	M	M	M
EA Gross type	C (t)		C		C (t)		C (t)		C (t)		C	

**Table 2 genes-14-01822-t002:** Hypermethylated genes involved in Rho GTPase pathway.

Symbol	Gene Name	OMIM
*ARHGAP36*	Rho GTPase activating protein 36	300937
*ARHGAP4*	Rho GTPase activating protein 4	300023
*ARHGAP6*	Rho GTPase activating protein 6	300118
*ARHGEF6*	Rac/Cdc42 guanine nucleotide exchange factor 6	300267
*ARHGEF9*	Cdc42 guanine nucleotide exchange factor 9	300429
*FGD1*	FYVE, RhoGEF, and PH domain containing 1	300546
*GDI1*	GDP dissociation inhibitor 1	300104
*MCF2*	MCF.2-cell-line-derived transforming sequence	311030
*OCRL*	OCRL inositol polyphosphate-5-phosphatase	300535
*STARD8*	StAR-related lipid transfer domain containing 8	300689

## Data Availability

Data is available upon request.

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
