# Peer review of "Epigenetic Findings in Twins with Esophageal Atresia"

_genes, 2023, doi:10.3390/genes14091822_

Round 1

Reviewer 1 Report

Michal et al. apply RRBS of blood and esophageal tissues to identify epigenetic factors in ethiopathogenesis of esophageal atresia. Six pairs of twins are included, and pathway enrichment analysis reveals the involvement of 10 hypermethylated genes in the Rho GTPase pathway, which has not been previously described in the pathogenesis of EA. It is interesting to study the methylation change of EA. However, I have some concerns.

1. This study aims to study ethiopathogenesis of EA by analyzing RRBS data of 6 pairs of twins, but only involved 4 esophagus samples from EA patients. The main discovery, such as the 10 hypermethylated genes in the Rho GTPase pathway, are identified from the whole-blood samples of participants. Do whole blood cells take part in the pathogenesis of EA? The authors need to explain the relationship between the differences in whole-blood methylation levels and EA. If it is hard to collect esophagus samples, maybe analysis of plasma cell-free DNA methylation could provide additional information.

2. Since EA has higher incidence in males than females and age could affect methylation levels, general information of the included twins, such as sex and age, should be provided. These 10 hypermethylated genes (Table 1) are all located on the X chromosome. Whether these genes are related to EA, or is it a biased result caused by the unbalanced sex ratio of cases and controls? Moreover, the sample size is too small to draw valid conclusions.

3. It is recommended that the authors incorporate omics data to interpret hypermethylation and hypomethylation signals.

4. Gene symbols should be italicized.

5. Details of sample preparation and sequencing, such as DNA input amount, sequencing type (paired-end or single-end) and depth, need to be described in Methods.

The English language is appropriate and understandable.

Author Response

  1. This study aims to study ethiopathogenesis of EA by analyzing RRBS data of 6 pairs of twins, but only involved 4 esophagus samples from EA patients. The main discovery, such as the 10 hypermethylated genes in the Rho GTPase pathway, are identified from the whole-blood samples of participants. Do whole blood cells take part in the pathogenesis of EA? The authors need to explain the relationship between the differences in whole-blood methylation levels and EA. If it is hard to collect esophagus samples, maybe analysis of plasma cell-free DNA methylation could provide additional information.

Thank you for this remark. In revised version we included in discussion section links of Rho GTPase pathwayto immune regulation (based on very recent review by Rana El Masri and Jérôme Delon [PMID: 33547421]). This link, in turn, looks stronger in context of our whole-blood based study. However, we are very cautious at drawing strongly sounded conclusions.

 In addition we suggested use of plasma cell-free DNA methylation measurements to overcome limited access to esophagus samples.

  1. Since EA has higher incidence in males than females and age could affect methylation levels, general information of the included twins, such as sex and age, should be provided. These 10 hypermethylated genes (Table 1) are all located on the X chromosome. Whether these genes are related to EA, or is it a biased result caused by the unbalanced sex ratio of cases and controls? Moreover, the sample size is too small to draw valid conclusions.

We agree with Reviewer. However, our statistics (linear regression) was adjusted to gender (This has been noted in „Differential methylation (DM) analysis” section).  In addition we included in our results hierarchical clustering analysis with and without X and Y chromosomes (figures 3 and 5, respectively). The grouping of samples was similar indicating that gender balance has little influence on our data.

We are aware that we studied small sample size. Both limitations has already been placed in discussion.

  1. It is recommended that the authors incorporate omics data to interpret hypermethylation and hypomethylation signals.

The  solution here is to provide parallel transciptome analysis. Which is not possible right now due to financial limitations. This has already been noted in discussion.

  1. Gene symbols should be italicized

We agree with Reviewer.

  1. Details of sample preparation and sequencing, such as DNA input amount, sequencing type (paired-end or single-end) and depth, need to be described in Methods.

DNA has been paired-end sequenced (2×100bp) on HiSeq 2500 Sequencer (Illumina) in six batches. - we've added this information in methods.

Reviewer 2 Report

Interesting and well-built study on the role of epigenetic factors in esophageal atresia, highlighting the statistically significant involvement of 10 hypermethylated genes in the onset of esophageal atresia.

Author Response

Tank you for your review.

Reviewer 3 Report

This is an interesting paper that wants to study epigenetic factors in determinism of AE in twins,

Line 21: change order, in the blankets the short word: “The RRBS technique (Reduced-Representation Bisulfite Sequencing)”

Line 33: use “tracheo-esophageal”  for “tracheoesophageal”

Line 39: it is better “syndrome”

Line 41: write better “So far, over 50 diseases and syndromes in which EA is one of the symptoms have been described”

Line 54: why “processes – differentiation”

Line 56: four models? You write three

Line 61 again “postulated – individual”

Line 85: again “RRBS (Reduced Representation Bisulphite Sequencing)”

The results are too difficult to understand, try to change

Author Response

Line 21: change order, in the blankets the short word: “The RRBS technique (Reduced-Representation Bisulfite Sequencing)”

The Reduced-Representation Bisulfite Sequencing technique (RRBS)

Line 33: use “tracheo-esophageal”  for “tracheoesophageal”

tracheo-oesophageal or tracheoesophageal, we use esophagus instead of oesophagus in article, so we prefer tracheoesoephageal.

Line 39: it is better “syndrome”

agree

Line 41: write better “So far, over 50 diseases and syndromes in which EA is one of the symptoms have been described”

EA is one of the symptoms found in over 50 different diseases and syndromes.

  Line 54: why “processes – differentiation” The course of embryogenesis processes differentiation of the trachea and esophagus (...)  

The course of embryogenesis processes differentiation of the trachea and esophagus from foregut is the subject of research conducted on animal embryos [12].

Line 56: four models? You write three outgrowth model, watershed model, septation model, splitting and elongation model  

Four models of how the trachea emerges from the foregut have been described already: outgrowth model, watershed model, septation model, splitting and elongation model [12,13].

Line 61 again “postulated – individual” a multifactorial etiology is postulated, individual sensitivity and the interaction of genetic and environmental factors leading to EA [14].  

Due to the low risk (population risk) of repeating the isolated form of EA in another child and the low risk of EA in the next generation, a multifactorial etiology is postulated, individual sensitivity and the interaction of genetic and environmental factors leading to EA [14].

Line 85: again “RRBS (Reduced Representation Bisulphite Sequencing)” (...) by use of Reduced Representation Bisulphite Sequencing technique (RRBS).

Genome wide methylation has been studied by use of Reduced Representation Bisulphite Sequencing technique (RRBS).   The results are too difficult to understand, try to change
We analysed our data with most recent version of various software dedicated to Reduced Representation Bisulphite Sequencing technique (RRBS). We did our best to display our results logically. Therefore, if Reviewer thinks it is hard to follow we need more guidance/hints to improve our manuscript in this context.

Round 2

Reviewer 1 Report

Although there is some improvement, essential major points (#1 and #2) I proposed have not been fully explained. In addition, the revised manuscript is still premature to be published due to the weak link between the methylation changes of whole-blood samples and ethiopathogenesis of EA.

The following are some comments for further improvement.

1. The authors observed that 10 genes enriched in Rho GTPase pathway were hypermethylated in whole blood samples from EA patients. Genes involved in Rho GTPase pathway play multiple roles in tissues including immune cells, and the authors claimed that “involvement of Rho GTPases in immune regulation may suggest that pathogenesis of EA may have been linked to abberrant inflammatory responses” (lines 342-344). However, it is also possible that the methylation changes is linked to the effects of EA. The authors failed to provide sufficient evidence linking methylation changes to the pathogenesis of EA.

2. The similarity of hierarchical clustering analysis with and without X and Y chromosomes (figures 3 and 5) is insufficient to rule out the influence of X chromosome, as most CpG islands are from autosomes.

3. Box plots are recommended to visualize the DNA methylation profiles of the 10 genes in Rho GTPase pathway.

4. It would be better to include clinical features of the patients using images if possible.

5. Data processing is not part of Statistical analyses and should be divided into separate sections.

6. Line 253, Figure 6 should be Figure 7

7. Line 344, a typo, “abberrant”

The English language is appropriate and understandable, but there are some careless mistakes.

Author Response

1. The authors observed that 10 genes enriched in Rho GTPase pathway were hypermethylated in whole blood samples from EA patients. Genes involved in Rho GTPase pathway play multiple roles in tissues including immune cells, and the authors claimed that “involvement of Rho GTPases in immune regulation may suggest that pathogenesis of EA may have been linked to abberrant inflammatory responses” (lines 342-344). However, it is also possible that the methylation changes is linked to the effects of EA. The authors failed to provide sufficient evidence linking methylation changes to the pathogenesis of EA.

The reviewer is right. We suggest that pathogenesis of EA may have been linked to abberrant inflammatory response, but it is also possible that the methylation changes is linked to the effects of EA. We add the sentence to the main text after statement „involvement of Rho GTPases in immune regulation may suggest that pathogenesis of EA may have been linked to abberrant inflammatory responses"  such as: „but it is also possible that the methylation changes is linked to the effects of EA and further research is needed to obtain more evidence for this hypothesis.”

2. The similarity of hierarchical clustering analysis with and without X and Y chromosomes (figures 3 and 5) is insufficient to rule out the influence of X chromosome, as most CpG islands are from autosomes.

The reviewer is right, we suggest that further study on the bigger group of a patients is needed.

3. Box plots are recommended to visualize the DNA methylation profiles of the 10 genes in Rho GTPase pathway.

We add box plots of 10 genes in Rho GTPase pathway as Figure 8.

4. It would be better to include clinical features of the patients using images if possible.

We add Table 1 with details about each twin pair: gender, zygosity, Gross type of EA.

5. Data processing is not part of Statistical analyses and should be divided into separate sections.

The reviewer is right, now it is section 2.3. Data processing and 2.4 Statistical analyses.

6. Line 253, Figure 6 should be Figure 7

The reviewer is right.

Line 344, a typo, “abberrant”

The rewiwever is right.

Reviewer 3 Report

good

Author Response

Thank you for your review.

Round 3

Reviewer 1 Report

It is interesting to study epigenetic factors in EA, and this study included twin samples, which are very precious. Although the authors have replied to each comment during the two rounds of revision, the fundamental problem has not been solved. As they described in the introduction, they plan to study the pathogenesis of EA. If this is the case, at least disease-related tissue samples need to be analyzed. The authors compared methylation signals of blood samples from EA patients and controls without clearly explaining whether and how blood cells participate in the pathogenic process of EA. Actually, whole blood provides biomarkers for diseases, but these biomarkers can only indicate a certain correlation with the disease. Without functional analysis, the differential methylation signals from whole blood cannot be linked to the pathogenesis of EA.

If the authors aim to study the pathogenesis of EA, they should compare esophageal tissue samples from EA patients and controls. If the authors aim to identify biomarkers of EA, the current design could support the research. However, the manuscript should be reorganized especially the title, abstract, introduction, and discussion sections to avoid misunderstandings.

Author Response

We have changed the title "Epigenetic findings in twins with esophageal atresia."

We have made appropriate changes to avoid suggesting a direct relationship between the presented results and the pathogenesis of esophageal atresia
